# Multi-Omics Analysis of Gut Microbiota in Inflammatory Bowel Diseases: What Benefits for Diagnostic, Prognostic and Therapeutic Tools?

**DOI:** 10.3390/ijms222011255

**Published:** 2021-10-19

**Authors:** Vickie Lacroix, Alexis Cassard, Emmanuel Mas, Frederick Barreau

**Affiliations:** 1Gastroenterology, Hepatology, Nutrition, Diabetology and Hereditary Metabolic Diseases Unit, Children’s Hospital, CHU de Toulouse, 31300 Toulouse, France; vickie.lacroix@inserm.fr; 2IRSD, University of Toulouse, INSERM, INRAE, ENVT, UPS, 31024 Toulouse, France; 3Department of Nephrology and Organ Transplantation, Rangueil Hospital, CHU de Toulouse, 31300 Toulouse, France; cassard.a@chu-toulouse.fr; 4Gastroenterology, Hepatology, Nutrition, Diabetology and Hereditary Metabolic Diseases Unit 330, Avenue de Grande-Bretagne, TSA 70034, CEDEX 09, 31059 Toulouse, France

**Keywords:** inflammatory bowel disease, Crohn’s disease, ulcerative colitis, microbiota, omics

## Abstract

Inflammatory bowel diseases (IBDs), which include Crohn’s disease and ulcerative colitis, are multifactorial diseases that involve in particular a modification of the gut microbiota, known as dysbiosis. The initial sets of metataxonomic and metagenomic data first made it possible to approximate the microbiota profile in IBD. In addition, today the new ‘omics’ techniques have enabled us to draw up a functional and integrative map of the microbiota. The key concern in IBD is to develop biomarkers that allow us to assess the activity of the disease and predict the complications and progression, while also guiding the therapeutic care so as to develop personalized medicine. In this review, we present all of the latest discoveries on the microbiota provided by “omics” and we outline the benefits of these techniques in developing new diagnostic, prognostic and therapeutic tools.

## 1. Introduction

Inflammatory bowel diseases (IBD), of which the two most common forms are Crohn’s disease (CD) and ulcerative colitis (UC), affect more than 3.5 million individuals across the world today [1]. The physiopathology of IBD results from an exaggerated immune response to an altered microbiota known as dysbiosis [2]. This abnormal ‘crosstalk’ is favored by an increase in the intestinal permeability [3]. One of the determining factors involved in the genesis of IBD and in the development of intestinal inflammation is the gut microbiota. So far, the exact mechanisms associated with the development of dysbiosis are however still unknown. The luminal load of bacteria is described to reach more than 10^14^/g of colonic content. The microbiota is established early in life and becomes stable in the first 2 to 3 years of life. This dynamic ecosystem does nevertheless present a functional stability, which allows it to perform its main functions, namely: acting as a barrier by limiting the establishment of pathogenic bacteria (competition for nutrients and synthesis of antimicrobial peptides), nutrition (metabolization and synthesis of some nutrients essential in humans etc.) and developing and maturing the immune system. Characterizing the biodiversity of the microbiota has long been limited by the cell culture techniques, and by a lack of access to virome, mycobiome and archaeome studies. More recently, the development of new high throughput sequencing techniques, the ‘omics’, has led to the discovery of the immensity of the microbiota and to the development of new strategies to study this latter and its involvement in IBD. These “multi-omics” technologies should be more easily available in the close future for humans. Thus, a precise characterization of the human intestinal microbiota and its functions may provide valuable assistance on the diagnosis, follow-up, prevention of relapses or complications and treatment of IBD.

## 2. Characteristics of Gut Microbiota in IBD

### 2.1. Metataxonomic and Metagenomic Data

The initial metataxonomic data, that take into account the bacterial diversity making up the microbiota, were obtained by amplifying and sequencing 16S rRNA which is highly conserved across species [4]. The sequences were merged into phylotypes or operational taxonomic units (OTUs). They show a significant reduction in the diversity of the microbiota in both UC and CD [5]. Ott et al. used a method of single-strand RNA conformation polymorphism on colonic biopsies. They report a 50% reduction in the microbial diversity in CD patients and a 30% reduction in UC patients [5]. The main genera found to be reduced in faecal samples of CD patients were *Bacteroides*, *Eubacterium*, *Faecalibacterium and Ruminococcus* (Figure 1). Among these, the reduction mainly concerns the butyrate-producing species, which are essential to the energy metabolism of the intestinal epithelium (*Roseburia inulinivorans, Ruminococcus torques, Clostridium lavalense, Bacteroides uniformis, Faecalibacterium prausnitzii*) [6,7,8].

The first metagenomic approaches used to sequence all of the 16S rRNA sequences of stool from healthy individuals and CD patients confirmed the reduction of *Firmicutes* [9]. Joossens et al. thus identified five bacterial species that are characteristic of the dysbiosis associated with CD. They observed a reduction in *F. prausnitzii*, the most specific element [10,11], in *Bifidobacterium adolescentis*, in *Dialister invisus*, and in a species belonging to the *Clostridium* XIVa cluster, and finally an increase in *Ruminococcus gnavus* [12]. An alteration in gut biodiversity is also present in UC, although this has been less studied. Machiels et al.’s quantitative taxonomic analysis by RT-PCR on the faeces of a cohort of 127 UC patients suggested a significant reduction in butyrate-producing bacteria as previously described. However, the most specific parameter of dysbiosis in UC patients appears to be the reduction of *Roseburia hominis* [11]. Additionally, a more recent meta-analysis, merging in silico data from five cohorts from developed countries, found an increase in *F. prausnitzii* in UC patients compared to healthy control groups, suggesting an adaptive mechanism [13]. The dysbiosis noted in IBD thus appears marked. Studies using the Bray–Curtis dissimilarity (which assesses the similarity in abundance of bacterial species between two samples) to compare beta diversity reported a significant difference that persists over time between healthy microbiota and IBD microbiota [1,8,13], which is more marked in CD (notably with ileocecal resection) than in UC patients [8].

The reduction in commensal bacteria diversity is linked to an increase in pathogenic species. In IBD, *Enterobacteriaceae*, notably adherent-invasive *Escherichia coli* (AIEC) are increased [14]. They were observed in 21.7% of biopsies of chronic ileal lesions from CD patients compared to 6.2% in the control group [15]. Among others, the following were found in UC patients: an increase in the *Desulfovibrio* load on colonic biopsies [16], an increase in pathogenic bacteria invading the epithelium such as *Fusobacterium varium* [17].

Several longitudinal studies have indeed suggested that the microbiota of IBD patients presents significant intra-individual instability over time [1,8,9,18]. Halfvarson et al. used PCR techniques to compare the V4 16S rRNA variable region on 3-monthly faeces samples in 137 IBD patients [8]. They found a variability in the alpha diversity that was significantly greater in IBD patients, particularly for CD patients after ileo-caecal resection [8]. Lloyd et al. reported the general instability rate of the IBD microbiota taxonomic profile to be slightly increased compared to the control group. However, the main species concerned (*P. copri*, *F. prausnitzii*, *E. coli*) that account for 40% of this rate, differed significantly [1].

### 2.2. Limitations of These Techniques in Studying the Dysbiosis Associated with IBD

Metataxonomic and metagenomic techniques do have some limitations, which raise questions as to the interpretation of the data obtained up until now. “Microbial dark matter” refers to unknown and under-characterized microbiotic biomass, where the metagenomic profile cannot currently be assigned in the classification of the microbial kingdom [19]. Indeed, less than 1% of bacterial and archaeal species can be obtained in culture [20]. The pathophysiological role in IBD of such matter is thus underestimated. Recently, Almeida et al. performed an in silico reconstruction of 92.143 genomes from thousands of metagenomes taken from 75 studies on human microbiota. They evidenced 1952 non-cultivable species of which 74% were not previously known, thus increasing the phylogenetic diversity of the microbiota by 281%. Thus, they suggested roles in intestinal metabolic pathways that were distinct from the referenced cultivable microbiota [21].

Finally, predicting the functional profile of the abnormal microbiota in IBD simply by the presence or absence of some specific genes sometimes appears to be at odds with the real activity of the bacteria from a transcriptomic, proteomic and metabolomic point of view. By integrating several omics techniques, Heintz–Buschart et al. noted a weak correlation between metagenomics and metatranscriptomics [22]. For their part, Lloyd et al. showed that the difference in the metabolomic profile between IBD patients and control groups, independently of both disease activity and dysbiosis scores, was more significant than that of the taxonomic profile [1].

## 3. Contribution of Multi-Omics Techniques in Studying the Microbiota

### 3.1. Transcriptomics, Proteomics and Metabolomics

The emergence of new high throughput sequencing technologies, known as “omics”, now provides a better understanding of the functional impact of dysbiosis in IBD [1,8,19]. Firstly, metatranscriptomics made it possible to dynamically assess the gene expression profile of the microbiota using RNA-seq techniques associating the isolation of total RNA then the depletion in mRNA from the host [23]. A comparative metagenomic/metatranscriptomic analysis of 78 faecal samples performed by the integrative Human Microbiome Project (iHMP) showed an absence of a systematic correlation between the abundance of a genome and its functional activity. Indeed, Schirmer et al. demonstrated an absence of transcriptional activity of *D. invisus* while its DNA was quantitatively high in IBD patients’ faeces. In the same way, a high degree of variation was found between DNA and RNA quantities of *F. prausnitzii* [24]. Thus, multiple metabolic pathways involved in IBD through their role in inflammation or immune response had a modified expression, such as the dTDP (deoxythymidine diphosphate)-L-rhamnose biosynthesis pathway by *F. prausnitzii* and the methylerythritol phosphate (MEP) pathway by *A. putredinis* [24].

Subsequently, combining metagenomic and metaproteomic data has played a key role in characterizing signaling proteins and pathways, which were previously unknown but involved in the pathogenesis of IBD. Using two-dimensional liquid chromatography techniques alongside tandem mass spectrometry (2d-LC-MS/MS), Erickson et al. identified 116 protein clusters whose expression was significantly different depending on the patients’ phenotype. They also evidenced a wide diversity of proteins whose functions were previously unknown (29 to 31% depending on the study techniques). A reduction in the abundance of proteins involved in metabolic functions, energy production, defence and intracellular signaling was also reported in CD patients [25,26]. The dysbiosis is responsible for an increase in the expression of surface proteins in gram-negative bacteria (TonB, OmpA, RagB, SusC/D) [27], of which the majority play an antigenic role, and can therefore contribute to the abnormal amplification of the immune response. Additionally, there is a loss of proteins involved in the production of Short-Chain Fatty Acids (SCFAs) and in the degradation of mucins [26].

Finally, the recent contribution of metabolomics is not negligible since this makes it possible to analyze the metabolic profiles in various samples. It uses techniques of nuclear magnetic resonance (NMR) spectroscopy, of reversed-phase liquid chromatography or gas chromatography coupled with mass spectrometry (respectively LC-MS and GC-MS) combined with a bioinformatic tools. Santoru et al. analyzed the microbiota of 183 subjects and demonstrated a significant difference in the concentration of metabolites between IBD patients and healthy controls. A strong association was demonstrated by Spearman’s rank correlation between five bacterial types (*Oscillospira*, *Faecalibacterium*, *Escherichia*, *Flavobacterium* and *Veillonella*) and 10 metabolites (biogenic amines, amino acids, lipids and vitamins), which was greater in CD patients. For example, two bacterial amines, cadaverine and putrescine, whose catabolism is responsible for an increase in oxidative stress, were shown in greater quantities in CD patients, and were negatively correlated to the levels of two bacteria belonging to the *Firmicutes* phylum, *Faecalibacterium* and *Oscillospira* [28]. Using NMR, Marchesi et al. also reported a reduction of SCFAs in faeces of CD patients. Complementary analyses showed a reduction in members of the *Clostridium coccoides* and *Clostridium leptum* groups, which are partly responsible for the production of SCFAs [29]. A depletion in vitamins B3 and B5 was also observed in CD patients. These vitamins are involved in the production of cofactors of the lipid metabolism and in the protection of the intestinal mucosa. This reduction was correlated to the reduction in *F. prausnitzii*, which produces nicotinic acid [1,28]. Lloyd-Price et al. confirmed the increase in primary bile acids (cholate) and the reduction in secondary bile acids (lithocholate and deoxycholate) in faecal samples of IBD patients, suggesting an alteration in the production of these latter [1].

### 3.2. Limits of Multi-Omics Techniques

These techniques do however have limitations. The samples can be contaminated by cells from the host. Thus, Lehmann et al. proposed using a more sensitive mass spectrometry technique or a human protein depletion in order to overcome this bias [25]. Transcriptomics have allowed us to discover many genes, but their functions are still unknown. Thus, it must be coupled with other techniques. For its part, proteomics does not make it possible to study the mechanisms responsible for protein alteration (reduction in the expression, increase in the degradation, reduction in the microbial diversity). These ‘multi-omics’ data are now included in several international databases such as the Human Microbiome Project (HMP), Human Proteome Project (HPP) and Human Metabolome Database (HMDB). However, they require a common interpretation to allow an integrative point of view of how the different components of the microbiota may contribute to IBD pathogenesis [19].

### 3.3. Integrative Mapping of Multi-Omics Data

Lloyd-Price’s recent work has made it possible to highlight the interactions between the longitudinal data provided by the five omics (metataxonomics, metagenomics, metatranscriptomics, metaproteomics, metabolomics) cited in IBD, using an in silico construction of a large-scale integrative map [1]. Among the 300 most significant correlations, there is the reduction of *F. prausnitzii* and simultaneous increase of *E. coli;* the association between the reduction of *Roseburia* and the dysregulation of both acylcarnitine and bile acid metabolisms; the central role of the abundance of C8 carnitine, which, in association with the increase in cholate and its derivatives, represents 6% of the associations. Several genes from the host are revealed to be highly significant hubs in connection with the dysbiosis, such as for example the gene coding for glucose-dependent insulinotropic polypeptide, and participating in the postprandial gastrointestinal motility. An overexpression of this gene could thus contribute to increase the intestinal peristalsis and the digestive disorders and in CD [30]. The authors also noted an ileal overexpression of RNA polymerase, suggesting an increased growth of the microbiota in a condition of dysbiosis.

Although it seems essential to associate these omics data for a holistic vision of the microbiota-host interactions in the pathophysiology of IBD, this mapping does not provide information about the direction of these relationships. Is this the influence of the microbiota on the host, or the other way round, or is it a reciprocal impact? In the inflammatory epithelial regions of CD, there is an overexpression of genes coding for proteins involved in anti-bacterial defences such as chemokine C-X-C motif ligand 6 (CXCL6), or dual oxidase 2 (DUOX2), which catalyzes the synthesis of reactive oxygen species, which are mediators of the inflammation. A search for covariation of the host’s metatranscriptome and the microbiota’s metagenome has revealed a negative correlation of DUOX2 with *Ruminococcaceae* and of CXCL6 with *Eubacterium rectal* in the CD ileum [1]. Although it is possible that the expression of these genes in the host can modify the microbiota, there are no data currently available to ascertain the direction of the relationship that links them.

## 4. Implications of the Multi-Omics Approach in Studying IBD

### 4.1. The Microbiota, a New Biomarker for IBD

More recently, some studies have sought to determine the predictive value of these differences in microbiota in terms of relapse of the disease. On faecal samples taken when stopping Infliximab treatment, then at 2, 6 and 18 months, in 33 CD children, Rajca et al. compared the taxonomic profiles of the patients in clinical remission with those of the relapsed patients [31]. They suggested that a low level of *F. prausnitzii* and of *Bacteroides* at diagnosis are predictors for clinical relapses. Sokol et al. also demonstrated a correlation between the reduction of *F. prausnitzii* and a higher risk of relapse after ileal resection surgery [32]. More recently, he also compared the ileal microbiota of 201 patients by 16S rRNA sequencing at the time of their ileocecal resection and over the following year. He reported that the abundance of other taxa at the time of the resection was significantly associated with a risk of endoscopic relapses, including *Gammaproteobacteria*, *Corynebacterium* and *Ruminococcus gnavus* [33]. Thus, assessing the abundance of these taxa at the time of surgery appears to be a powerful predictive tool for relapse, with areas under the Receiver Operating Characteristic (ROC) curve in a random forest model of 81% (60.8–100%). A prospective paediatric cohort looked at finding microbiotic biomarkers associated with a risk of complications. Kugathasan et al. followed a cohort of 913 children with CD and showed a correlation between the abundance of *Rothia* and *Ruminococcus* and the appearance of stenosis, as well as between *Collinsella* and the appearance of fistulas [34]. It should be noted that the impact of these data on prediction risk has not been assessed.

### 4.2. Using Omics Techniques to Assess Treatment Response

Assessing disease activity and treatment efficacy by studying the microbiota has also been considered. Thus, Lewis et al. used metagenomics on 86 faecal samples from CD children to assess the response to enteral feeding treatment or anti-TNF (tumour necrosis factor) [35], a good response being associated with a reduction in *Actinomyces* and an increase in *Lactococcus* and *Roseburia* [36]. Sanchis-Artero et al. are planning a quantitative and qualitative analysis of the microbiota by Illumina sequencing techniques at inclusion and 6 months later, of 88 subjects with CD starting an anti-TNF treatment and a control group, in order to study the use of *F.prausnitzii/E. coli* and *F.prausnitzii/C. coccoides* ratios as therapeutic indicators [37].

### 4.3. Dysbiosis, a Therapeutic Target in the Treatment of IBD

The gut microbiota thus appears to be a promising target in therapeutic care for IBD patients. Faecal microbiota transplant (FMT) is a recent treatment, consisting of directly transferring gut microbiota from a healthy donor into the recipient patient’s digestive tract. This technique has already been proven to be effective in refractory *Clostridium difficile* infections (85–90% success rate), and now seems to be a promising technique in treating IBD. Fang et al.’s meta-analysis pooled a total of 23 cohort studies and four randomized clinical trials for UC to assess clinical remission after FMT [38]. In total, 21% of UC patients and 30% of CD patients achieved clinical remission after FMT. In a second meta-analysis by Paramsothy et al., post-FMT analysis of the microbiota demonstrated an increase in the microbial diversity in the recipients, whose taxonomic profiles moved closer to those of the donors [39]. However, the lack of data, the heterogeneity of pathologies and implementation protocols and the lack of randomized clinical trials make it difficult to interpret these results, which explains the absence of an indication for FMT in the treatment of IBD [40,41]. Indeed, the host’s genetics can also influence the effects of FMT: the *NOD2* (Nucleotide binding oligomerization domain 2) mutations found in CD and known for its role in dysbiosis, could eventually reverse the effects of the FMT. Indeed, the dysbiosis induced by the deletion or mutation of the *Nod2* gene has been described as transmissible and dominant [42]. In this article, the authors used an approach using embryo transfer (transfer of embryos expressing or not expressing *Nod2* in wild type mice) to demonstrate that the embryos not expressing *Nod2* and receiving wild type microbiota developed a dysbiotic flora over time comparable to mice not expressing *Nod2* and receiving dysbiotic microbiota [42]. Finally, the dysbiotic microbiota of these mice not expressing *Nod2* ended up being established by coprophagia in the mice expressing *Nod2* who shared the same cage (dizygotic twins; transmissible and dominant features of this dysbiotic flora). This dysbiotic flora participates in the harmful effects of the *Nod2* gene on the homeostasis of the intestinal mucosa, the inflammation and the colonic carcinogenesis [42,43,44,45]. More recently, new therapeutic strategies are being studied that use genetically modified bacteria designed to produce different therapeutic substances delivered in situ (Live Biotherapeutic Products—LBPs). This involves constitutive or inducible systems, mostly using *Lactococcus lactis* and *E.coli* (for example, *L. lactis* that secretes interleukin-10 (IL-10), interleukin-27 (IL-27) or anti-TNF antibodies, *L. lactis* transfected with a plasmid producing microbial anti-inflammatory molecule (MAM) normally secreted by *F. prausnitzii*, *E. coli* that produces interleukin-35 (IL-35); *L. lactis* xylose inducible expression system (XIES) or *B. ovatus* which secretes TGF-β1 following a xylan-inducible system [46,47,48]. These live biotherapy products could represent new tools in the therapeutic arsenal for IBD.

### 4.4. Perspectives in IBD Management

Multi-omics thus pave the way for the development of new diagnostic, prognostic and therapeutic tools. Thereby, intestinal microbiota sequencing and multi-omics analyses could allow a best screening of IBD patients at diagnosis. First it would help to evaluate the risk of relapse, then to adjust the treatment (drug, dose) and propose a close supervision of high-risk patients.

They also fit into the idea of personalized medicine, where the therapeutic care, and particularly FMT, can be guided by a study of each patient’s gut microbiota. Identification of bacterial species associated to success or failure of TMF could help to refer the patients for FMT or not, and to predict the success rate, in order to propose other treatment (Figure 2). Another major challenge for the multi-omics approach is to develop useful tools from the bench to bedside.

## 5. Conclusions

Omics techniques have made a major contribution to characterizing dysbiosis in IBD. They enable dynamic studies over time as well as integrative studies that consider the different parameters obtained in gut microbiota studies. Several characteristics remain constant, such as the reduction of bacterial diversity affecting *Firmicutes*, notably *F. prausnitzii*, and the abundance of pathogenic germs such as adherent-invasive *E. coli*. It has thus been possible to detail the impact of these microbiotic modifications on the host by associating transcriptomic, proteomic and metabolomic data, which has provided information on the pathophysiological role of this dysbiosis. Thanks to multi-omic approach, microbiome-based biomarkers could be developed to evaluate the risk stratification, to predict disease evolution and treatment efficacy. This review provides a new insight in high throughput sequencing technologies to investigate microbiota of IBD patients and their contribution to clinical practice. There remain some difficulties to overcome in the development of robust biomarkers. Indeed, there is a great variability between studies in term of sampling (stool, biopsy) and criteria of selection for patients (disease activity, severity, therapeutic care). It is thus essential to homogenize these elements. Finally, omics tools are still expensive in current practice and it is difficult to interpret the immense quantity of data obtained.

## Figures and Tables

**Figure 1 ijms-22-11255-f001:**
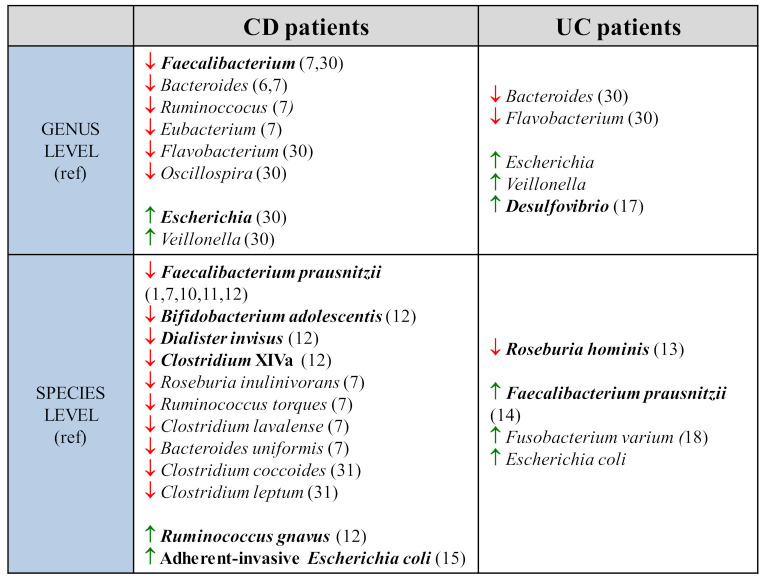
Taxonomic, metagenomic, metatranscriptomic and metabolomic data to evaluate microbiome composition in IBD (CD and UC patients). Comparison at genus and species levels, in CD and UC patients.

**Figure 2 ijms-22-11255-f002:**
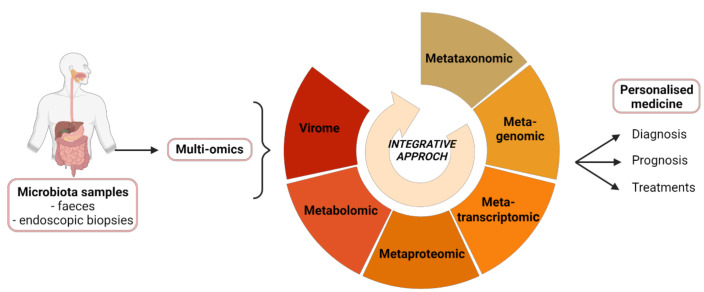
Integrative analysis of multi-omics data regarding the gut microbiota in IBD. The microbiome is characterized from stool samples or endoscopic biopsies, using multi-omics techniques. The provided data open up new horizons for the development of diagnostic, prognostic and therapeutic tools optimized for a personalized medicine. Created by BioRender.com.

## Data Availability

Not applicable.

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
