# Peer review of "Multi-Omics Analysis of Gut Microbiota in Inflammatory Bowel Diseases: What Benefits for Diagnostic, Prognostic and Therapeutic Tools?"

_ijms, 2021, doi:10.3390/ijms222011255_

Round 1

Reviewer 1 Report

I recommend to accept the manuscript after minor revision.

There are only some points to correct:

 - please provide the list of abbreviations

- introduction and discussion section need improvement; please provide information on how your results will translate into clinical practice

- in discussion section please provide study strong points  and study limitation section

- please correct typos

All abovementioned issues are crucial for the credibility of the results. The paper can be accepted only after addressing all the issues and another subsequent review.

I recommend to accept the manuscript after minor revision.

Author Response

Reviewer 1

I recommend to accept the manuscript after minor revision.

-“please provide the list of abbreviation”

In agreement we have provided a list of abbreviations.

-“Introduction and discussion section need improvement; please provide information on how your results will translate into clinical practice”

According to these comments, introduction and discussion sections have been modified.

-“in discussion section please provide study strong points  and study limitation section”

Thank you for this comment. However, we believe that such a presentation of limitation/strong points of each study would be confusing for the readers. Meanwhile, a special section of discussion presents the strong points for multi-omics from all the studies.

-“please correct typos”

We have checked and corrected the typos.

All abovementioned issues are crucial for the credibility of the results. The paper can be accepted only after addressing all the issues and another subsequent review. I recommend to accept the manuscript after minor revision.

Thank you for these comments.

Reviewer 2 Report

The authors reviewed and presented some interesting aspects on the possible role of multi-omics analysis for diagnostic, prognostic, and therapy in IBD. There are important new data on the advantages and limitations of these techniques.

The manuscript is well written, but there are few changes that I would recommend to be made.

  • The conclusions of this review must be reviewed and shortened, without reference to tables or other papers cited. Some sentences may be replaced before conclusions and be part of a separate paragraph regarding the idea of personalized medicine and the future perspective of the practical applicability of these tools.
  • Figure 2 may be redesigned as the list of Multi-omics probably would be better to be linked directly with the specified part of the integrative approach round.
  • The authors must revise the list of references for editing almost every reference according to the required template.

I consider these changes as minor and without altering the structure and the main ideas of this review.

Author Response

Reviewer 2

          The authors reviewed and presented some interesting aspects on the possible role of multi-omics analysis for diagnostic, prognostic, and therapy in IBD. There are important new data on the advantages and limitations of these techniques. The manuscript is well written, but there are few changes that I would recommend to be made.

          The conclusions of this review must be reviewed and shortened, without reference to tables or other papers cited. Some sentences may be replaced before conclusions and be part of a separate paragraph regarding the idea of personalized medicine and the future perspective of the practical applicability of these tools.

          We agree with these comments and we have modified the conclusion. Tus, the section of conclusion has been reduced and we have created a new section, before conclusion, entitled: Perspectives in IBD management

          Figure 2 may be redesigned as the list of Multi-omics probably would be better to be linked directly with the specified part of the integrative approach round.

          In agreement Figure 2 has been changed and the multi-omics has been integrated in the integrative approach round.

          The authors must revise the list of references for editing almost every reference according to the required template.

          In agreement, we have edited the references according the required template.

I consider these changes as minor and without altering the structure and the main ideas of this review.

Thank you for this comment.
